

# A novel approach to texture recognition combining deep learning orthogonal convolution with regional input features

Kar-Seng Loke

Industrial Management, National Taiwan University of Science and Technology, Taipei, Taiwan, Taiwan

## ABSTRACT

Textures provide a powerful segmentation and object detection cue. Recent research has shown that deep convolutional nets like Visual Geometry Group (VGG) and ResNet perform well in non-stationary texture datasets. Non-stationary textures have local structures that change from one region of the image to the other. This is consistent with the view that deep convolutional networks are good at detecting local microstructures disguised as textures. However, stationary textures are textures that have statistical properties that are constant or slow varying over the entire region are not well detected by deep convolutional networks. This research demonstrates that simple seven-layer convolutional networks can obtain better results than deep networks using a novel convolutional technique called orthogonal convolution with pre-calculated regional features using grey level co-occurrence matrix. We obtained an average of 8.5% improvement in accuracy in texture recognition on the Outex dataset over GoogleNet, ResNet, VGG and AlexNet.

Corresponding author
Kar-Seng Loke,
loke.ks@gapps.ntust.edu.tw

# INTRODUCTION

Textures are complex visual patterns that are characterized by the repetitive variation of smaller elements. These elements are subject to stochastic randomization, but the overall visual pattern is still distinctive. The variations can be in terms of element size, structure, orientation, color, intensity and so on. However, there is no complete and formal definition of texture yet. The local visual microstructure can have properties of uniformity, density, rough-ness, regularity, directionality, smoothness, granulation, homogeneity and so on. A region with constant texture is considered to have a set of local statistics or other local properties that remain constant or slowly varying within that region. Of course, the question is what are those properties and the answer to that that depends on the issues being addressed since there is no general agreement.

Textures can provide a powerful segmentation and object detection cue, so it is not surprising that the brain visual cortex areas encode them. There is evidence that neurons in the higher areas of the cortex from V2 (visual cortex area 2) encode naturalistic textures. Recent studies show that higher visual areas, such as V2, V3, and V4, respond more vigorously to images with naturalistic higher-order statistics than to images lacking

them (*Kim, Bair & Pasupathy, 2019*; *Movshon & Simoncelli, 2014*; *Freeman et al., 2013*). *Freeman et al. (2013)* have found that electrophysiological recordings of neurons in macaque that V2 and not V1 respond to higher-order statistical dependencies found in natural images. Since there are neurons that are specific to textures, texture recognition is fundamental in our visual systems. Therefore, we assume that artificial systems may also use textures in computer vision for object detection and recognition.

Similarly, it has been found that the higher (deeper) layers in a convolutional neural network (CNN) also encode naturalistic textures. *Laskar, Giraldo & Schwartz (2020)* found some correspondence in terms of response modulation to synthetic naturalistic texture btween V2 brain and higher CNN layers such as L2 (Layer 2) layers from AlexNet (*Krizhevsky, Sutskever & Hinton, 2017*) and L3 from VGGnet (*Simonyan & Zisserman, 2015*). Visualization of image features in the CNN revealed that layer 2 activation has some texture selectivity (*Krizhevsky, Sutskever & Hinton, 2017*; *Zeiler & Fergus, 2014*) that was not found in L1. This indicates that CNN L2 neurons are able to better separate the texture families than L1. L2 neurons are more selective to the high-order texture properties of different texture families. *Laskar, Giraldo & Schwartz (2020)* found that texture sensitivity first emerged in the second layer of AlexNet. However, they did not find similar sensitivity in the deeper layers beyond L2.

Similarly, *Zhuang et al. (2017)* found that CNN layers after L1 respond substantially better to textures than lower layers. These are consistent with findings in primates in that higher visual areas respond better to textures than V1. In the experiments of *Zhuang et al. (2017)*, they found that the response to texture is substantially higher than zero in deeper layers, *i.e.,* the population modulation index (PMI) is larger for layer 4 and 5 than layer 1, this implies that higher layer units preferentially respond better to texture images. The Population modulation index (PMI) measures the response of neuronal units to input stimuli as compared to noise.

Recent results also show that deep nets like ResNet (*He et al., 2016*) and GoogleNet (*Szegedy et al., 2015*) perform better than most using hand-crafted features such as local binary patterns (LBP), SIFT (Scale Invariant Feature Transform) and HOG (Histogram of Oriented Gradients). However, a shallower net Visual Geometry Group (VGG) (16 layers) performs even better than a deeper net like Googlenet (27 layers) and very close to ResNet on some datasets (*Khaldi, Aiadi & Kherfi, 2019*). CNN-based networks outperform all other handcrafted features in three of the non-stationary datasets. Non-stationary textures have local structures that change from one region of the image to the other. This is consistent with the view that CNN are good at detecting local microstructures disguised as textures.

Therefore, we should be mindful about the type of textures used. In *Freeman et al. (2013)*, *Zhuang et al. (2017)* and *Laskar, Giraldo & Schwartz (2020)* they used two types of synthetic textures: spectrally matched (SM) and correlation matched (CM) textures. SM texture is generated by keeping the (Fourier frequency) spectral energy constant while the phase is randomized. They had the same spectral properties as the original images. This, however, destroys the higher-order statistics, such as the correlations between linear filter responses in different scales of the original image (*Zhuang et al., 2017*). In the CM texture,

the mean, variance, skewness, and kurtosis of pixel intensity distribution are maintained. This basically includes the higher-order statistics from the pixels. We consider the mean to be first-order, variance 2nd-order, skew as 3rd-order and kurtosis as 4th order. The CM textures looked similar to the original images as judged by human observers.

On the other hand, stationary textures are textures that have statistical properties that are constant or slow varying over the entire region. These sorts of textures are not well detected by CNN. The gray-level co-occurrence matrix (GLCM) texture feature have better scores over CNN-based net with stationary texture datasets using 1 nearest neighbor classification (*Khaldi, Aiadi & Kherfi, 2019*).

Many hand-crafted feature approaches used a separate classifier such as K-nearest neighbour or support vector machine (SVM) because such features can't be input into a CNN. Moreover, a common view is that since CNN already extracts the features, there is no need to extract some prior features. The other assumption is that since deep nets already extract textures (*Geirhos et al., 2019*), there is no need for extraneous hand-crafted features. However, there are stationary and non-stationary texture (*Bello-Cerezo et al., 2019*) and there may be evidence that CNN may not cope well with stationary textures.

This research proposes the use of additional features apart from RGB values as integral input to the deep learning neural net. This will create neural activations that are not dependent on pixel RGB intensity convolutional feature maps. We will also introduce the novel use of a convolutional filter applied orthogonally to decouple the features spatially making it relatively more orderless. The basic idea of this research is that the use of RGB convolution is not sufficient to capture the type of statistics for non microstructure-based textures.

## MATERIALS & METHODS

### Related works

#### Features for texture

There are two major approaches to texture analysis, namely the structural approach and the statistical approach. The texton theory is representative of the structural approach. The basic idea is that textures are represented by primitive visual structures called texton. The term texton was introduced by *Julesz & Bergen (1983)* to describe pre-attentive texture elements. He called these elements "textons". These texton elements are elongated blobs (*e.g.*, rectangles, ellipses, or line segments) with specific properties, including color, angular orientation, width, length. However, as it lacked a precise definition, this approach did not lead to a computational development. The term texton was reintroduced by *Leung & Malik (2001)* to represent a finite vocabulary of image micro-structures which are small visual patterns captured by spatial filters.Therefore, a texton dictionary is obtained from a collection spatial filter bank of responses. The filter responses are clustered *via* k-means. Exemplar filter responses are chosen as textons and are collected into a dictionary. In the training stage, a histogram of textons, *i.e.,* the frequency with which each texton occurs in the texture is created for each the training image. In the classification stage, a histogram of textons for the input image is created and then compared with the rest of the input images.

A nearest neighbour classifier is used and the chi-squared statistic employed to calculate the similarity between the two probability distributions.

In contrast, a statistical approach attempts to determine the probabilistic distributions of the pixel intensities of the texture. It has been shown that second-order measures (statistics given by pairs of pixels) have achieved better discrimination rates than structural methods (*Materka & Strzelecki, 1998*). Second-order approaches would be based on the joint probability distributions of pairs of pixels. The most popular second-order statistical features for texture analysis are derived from the so-called gray level co-occurrence matrix (GLCM) (*Haralick, 1979*; *Haralick, Shanmugam & Dinstein, 1973*). Additional textural features are computed from the GLCM. These features include measures of angular second moment (energy) that measures homogeneity, inertia that measures contrast and entropy that measures randomness. The contrast feature is the measure of the amount of local variation in intensity. The correlation feature is a measure of linear dependencies of intensity within the region. *Haralick (1979)* defined 14 of such textural features.

Local binary patterns (LBP) is another popular approach. LBP (*Ojala, Pietikainen & Maenpaa, 2002*) capture spatial intensity patterns within a local window. Hence, they can be used to represent texture patterns. The pixels in the block are coded based on the difference in value of center pixel and its surrounding circular neighbors. For a $3 \times 3$ block, a chain code of 8 bit representing the circular neighbours is then obtained. The chain code bit is set to 1 if its intensity value is greater than the central pixel, otherwise zero. Hence, there are in total $2^8 = 256$ different patterns to describe the spatial relation of the center pixel. These bit patterns can be used to define local textural patterns.

To further illustrate, if T is the texture in a local neighbourhood, it is defined as a function t of the joint distribution of gray levels of P image pixels:

$$T = t(s(g_0 - g_c), s(g_1 - g_c), \ldots, s(g_{P-1} - g_c))$$

where $g_c$ represents the gray value of the center pixel of the local neighbourhood and $g_0, \ldots, g_{P-1}$ represents the surrounding neighbourhood pixels gray value. These neighbours form a circularly symmetric neighbourhood with radius R. And

$$s(x) = \begin{cases} 1, x \geq 0 \\ 0, x < 1 \end{cases} .$$

Then, $LBP_{P,R}$ represents a number that characterizes the spatial intensity structure of a local image region:

$$LBP_{P,R} = \sum_{p=0}^{P-1} s(g_p - g_c) 2^P .$$

In the training stage, an LBP histogram is built for each input image. In the classification phase, the LBP histogram of the input query image is obtained and compared. Many variations on this approach have been devised (*Liu et al., 2017*). In *Liu et al. (2016)* they concluded after testing various LBP approaches that the Median Robust Extended LBP

(MRELBP) gave the best results. The MRELBP uses the median value instead of the central pixel for thresholding.

In this section, we have briefly described the main traditional approaches to capturing texture features. The structural approach tries to describe textures as a combination of small visual patterns (microstructures) whereas statistical approach uses the distribution pattern of pixels.

### Deep learning approaches to texture

The popularity and advances in deep convolutional neural networks have led to an increase in research and exploration of such networks for texture recognition. A common approach is to use the convolution filter activation as features for use in classification. The input remains as the image RGB channels while the convolution filters are used for feature extraction. These outputs are used as textural features in various backend classification schemes.

*Andrearczyk & Whelan (2016)* introduced the T-CNN (T for texture). An important aspect of the architecture is the dual pathway architecture. It has a concatenation layer that combines "texture" and shape parts derived from separate layers of the network. The supposed texture components are derived from the pooled low-level features from the early convolution filter (*i.e.,* convolution layer C3). The pooling operation on each activation feature channel removes spatial information and therefore shape information. The more global presumed shape components are derived from later convolution layers (C5 layer). There are three fully connected layers after the concatenation layer. Their T-CNN-3 improved over AlexNet (*Krizhevsky, Sutskever & Hinton, 2017*) by 71.1% against 66.3% on the texture subset of imageNet (imageNet-T). On KTH-TIPS-2b, they obtained 48.7% over 47.6% against AlexNet.

*Cimpoi, Maji & Vedaldi (2015)* and *Cimpoi et al. (2016)* introduces FV-CNN as a texture descriptor that pools feature maps from the last CNN layer of the VGG network architecture. The pooling operation removes global spatial information and therefore is supposedly more appropriate for describing textures. It is called FV-CNN because it calculates features using the Fisher Vector from the pooling. To obtain the Fisher Vector (*Perronnin & Dance, 2007*; *Sanchez et al., 2013*), the local features from CNN activations are clustered using the Gaussian Mixture model. The Fisher Vector model includes an additional 1st and 2nd order moments so that a more precise description of the cluster assignment is obtained. This also reduces the number of clusters required for the same accuracy. They also included the FC-CNN features that captured the output of the penultimate fully connected layer of the CNN so that it can be used as features for the overall shape of the object. The final classification using the features encoded as Fisher vectors and FC features is performed using a support vector machine (SVM). These combined features obtained 81.1% accuracy on the KTH-T2b dataset using pre-trained VGG-19.

Instead of clustering, *Song et al. (2016)* used an ensemble of feedforward neural network on similar FC-CNN and FV-CNN descriptors to reduce their dimensions. The output from the fully connected neural network is used as the new transformed feature. They called this

network-based feature transformation. These transformed features are fed to a SVM for classification. They reported 83.3% accuracy on the KTH-TIPS2 dataset.

The Deep TEN texture encoding network (*Zhang, Xue & Dana, 2017*) also obtained their features from the CNN activations to create a set of dictionary words. Instead of using Fisher Vectors, they introduced a learnable residual encoding layer. The residuals are calculated as the difference between visual descriptors obtained by the CNN activations of the input and the dictionary codewords. This is similar to the second moments in the Fisher Vectors. Their encoding layer is a generalization of other similar residual encoders such as the popular vector of locally aggregated descriptors (VLAD) and Fisher Vectors. They obtained 82.0% accuracy on KTH-TIPS-2b dataset (11 texture categories with 4 samples per category), and improvement over FV-CNN which obtained 71.0% accuracy. *Liu et al. (2016)* concluded that that FV-CNN provides better results compared to LBP especially with textures that have a large within class variance in appearance after testing 27 methods on 14 datasets. They also concluded FV-CNN lack some robustness to noise and rotations.

To summarize, texture specific deep learning approaches generally utilize the CNN layers as features. The general approach is to pool the CNN features and create a dictionary encoding that use some kind of representation like Fisher Vectors, VLAD (*Arandjelovic et al., 2016*) or residual encoding layer. Apart from average pooling, pooling can also be achieved *via* a bilinear method as in *Lin & Maji (2016)* approach with BCNN. Also usually following the pooling and encoding, a classifier is employed, and this could be a support vector machine (SVM), k-nearest neighbors (K-NN) or a fully connected network. A texture representation and classification review article by *Liu et al. (2019)* presents further details on many of these approaches.

## Methods and approach

In this section, we describe the methods and dataset that we will be using. The input features and the network architecture are also described. Following that, we list the experiments that we have performed.

### *Texture dataset*

This study used the extended Outex texture that is released by University of Lille in France. The original Outex images are raster files of gray-level images. Most Outex test suites contain 24 different classes of textures, which generally yields very high classification results, therefore the new extended Outex database has expanded the number of textures to 68. Images have a set size of 128 × 128 pixels, each class has 20 training samples unless noted. Also, the illuminant is incandescent lighting unless stated otherwise. The resolution of the images is 100 dpi. Some of the classes in color Outex_TC_00013 are difficult to classify as they look visually similar. A summary of the Outex dataset is given in Table S1.

### *Feature representation*

*(A) Input layer representation.* The basic idea is that we want to represent a variety of pixel relationships that are not explicitly captured by the neurons. Convolutional neurons capture linear spatial pixel relationships as a dot product before the non-linearity is applied. In our approach the locations of each pixel in the spatial grid need not necessarily encode

the red-green-blue (RGB) values. For example, we may encode at each grid location a regional property or relational property that can be computed from pixels surrounding it. We could choose to encode each pixel a regional property centered at that pixel. Since each pixel location still preserve the spatial relationships, the convolutional neural layers will function appropriately, by capturing structural information within its receptive fields, except that instead of RGB values, they are capturing regional or relational properties.

*(B) Regional properties –GLCM and Haralick's features.* Ideally, we want properties that are informative and are not easily computed or approximated by the convolutional network at all layer levels. Therefore, any form of linear combination of pixel values is redundant. We should look for properties (or features) that are non-linear or of higher orders. One obvious regional property for texture recognition would be capturing textural properties at the region surround the pixel. Textural properties can be captured by grey level co-occurrence matrix (GLCM) or local binary patterns (LBP). We just have to expand the standard grayscale definition slightly to include color texture images.

GLCM calculates the global relational properties of neighbouring pixels globally as such it destroys the spatial grid arrangement of the pixels, and hence can't be used as an input layer to a CNN. One approach would be to calculate the GLCM on patches (instead of globally) surrounding the pixel in the grid. This would entail calculating on a sub-image instead of the entire image. One advantage of this is that the image size would be reduced and would reduce the computational burden for the CNN.

We can define image intensity as $I(x, y)$, where $(x, y)$ are the pixel coordinates. The image intensity are bounded by $0 \leq I(x, y) \leq 255$. We can create a multispectral co-occurrence matrix that counts the total number of pixel pairs in $I(x, y)$ having a value $I = i$ from the color channel a, and the value $I = j$ from another color channel b.

The pixel pairs are separated by a vector T where:

$$(x_2, y_2) = (t_x + x_1, t_y + y_1)$$

with $(x_1, y_1)$ as the coordinate of the first pixel, $(x_2, y_2)$ as the coordinate for the second pixel.

We define a co-occurrence matrix of colors a and b $(a, b \in \{R, G, B\})$ dealing with pixel pairs in $I(x, y)$ separated by a vector **T** as:

$$C_{abt} = \begin{bmatrix} c_{abt}^{00} & \cdots & c_{abt}^{0j} \\ \vdots & \ddots & \vdots \\ c_{abt}^{j0} & \cdots & c_{abt}^{ij} \end{bmatrix}$$

We then define $c_{abt}(i_a, j_b)$ as:

$$c_{abt}(i_a, j_b) = \sum_{x,y} \sum_{tx, ty \in U} \delta[I(x, y) - i] \times \delta[I(x + t_x, y + t_y) - j]$$

with $i_a$ and $j_b$ as intensity values from channels a and b respectively, **T** is the distance vector between the two pixels, $\delta$ the Kronecker Delta, and $x, y \in I$. Basically $i_a$ and $j_b$ are also the indices of the C matrix. The entire operation scans the image and for each $(x_1, y_1)$ - $(x_2, y_2)$

pairs records the intensity count according to the $i_a$ and $j_b$ column (which are the intensity value) in the C matrix. For example, $c_{abt}(i_a, j_b)$ for $i_a = 0$ and $j_b = 255$, a = R and b = G, is the count of all pixels at layer R with intensity 0 and at layer G with intensity 255. The pixels must be separated by a vector **T**. If $T = (t_x, t_y)$ is (0,1) then if $(x_1, y_1)$ is first pixel then $(x_2, y_2) = (x_1, y_1+1)$ is the second pixel. If the first pixel has intensity 0 and second pixel has intensity 255, then we add a count of one in $C_{RGT}$ matrix in column 0 and row 255. We accumulate the counts according to the conditions specified. This would create the GLCM.

Let, the set U be all possible $t_x$ and $t_y$ values that satisfy the condition $x^2 + y^2 = r^2$; r being a fixed distance from the center pixel, $r \in \mathbf{Z}$. In this case, the center pixel is the first pixel, and the second pixel is from the set of pixels $r$ radius away. These pixel pairs would be counted in the C matrix according to their intensities. For example, if the first pixel has 0 intensity and the second pixel $r$ radius away has intensity 255, a count of 1 is added to the column 0 and row 255 in the C matrix. The count is accumulated for all pixel pairs. This would yield a co-occurrence matrix with rotation-invariance. We would use this rotational invariant GLCM.

With 3 RGB layers, we obtain 6 multispectral matrices for each colour combination—RR, GG, BB, RG, RB and BG. Each of the six individual multispectral matrices, $C_{ab}$ (a, b $\in$ {R, G, B}) is normalized such that each matrix $C_{ab}(i, j)$ is $0 \leq i, j \leq 1$.

*(C) Sliding window calculation of GLCM patches.* We extract the GLCM patch from a sliding 5 × 5 window with a stride of 3 (Fig. S1). A 128 × 128 image (from Outext dataset) would give a 41 × 41 output. Within the 5 × 5 window we calculate a sliding 3 × 3 circular GLCM values for all the pixels within this window. The full GLCM for this window would be a 256 × 256 2D histogram. Since this value is quite large, we quantized the 256 values to either 16 or 64. This makes the GLCM size to be either 16 or 64.

From the quantized 2D-histogram, we calculate the 6 Haralick's features. For each window, we calculate contrast, inverse different moment, entropy, variance, correlation, second angular moment. We added the index of the largest value in the GLCM as another feature. This gives 7 features per 5 × 5 window.

We used the following features calculated on the GLCM matrices. Let

| | |
|---|---|
| $N$ | Number of distinct gray levels in the image |
| $p(i,j)$ | $(i,j)^{\text{th}}$ entry in the gray level co-occurrence matrix |
| $p_x(i)$ | $i^{\text{th}}$ marginal entry in the gray level co-occurrence matrix by summing rows of $p(i,j) = \sum_j^N p(i,j)$ |
| $p_y(j)$ | $\sum_i^N p(i,j)$ |
| $p_{x+y}(k)$ | $\sum_i^N \sum_j^N p(i,j), i+j = k, k = 2, 3, .., 2N$ |

Then, the textural features can be calculated as below:

Second Angular Moment
$$f_1 = \sum_i \sum_j p(i,j)^2$$

Contrast
$$f_2 = \sum_i \sum_j (i-j)^2 p(i,j)$$

Correlation
$$f_3 = \frac{\sum_i \sum_j (ij)p(i,j) - \mu_x \mu_y}{\sigma_x \sigma_y}$$

Variance
$$f_4 = \sum_i \sum_j (i - \mu)^2 p(i,j)$$

Inverse Different Moment
$$f_5 = \sum_i \sum_j \frac{1}{1 + (i-j)^2} p(i,j)$$

Entropy
$$f_6 = \sum_i \sum_j p(i,j) \log(p(i,j))$$

However, this only accounts for gray values. In order to consider colour values (RGB), instead of only gray values, we generate the GLCM histogram from cross color pairs—RG, RB and BG. For example, in RG, we take the central pixel from R layer and the surrounding pixel from the G layer. Since we calculate seven Haralick's feature and we have three cross color layer pairs, this would generate 21 values. This would be the input features.

We cast these 21 values as input features in a 41 × 41 grid basically as 41 × 41 × 21 tensor. So instead of the input as RGB values, they are now 21 Haralick's features. In addition, we can append the original RGB images to the input after resizing to match the size. Therefore, the input tensor would be 41 × 41 × 24.

We extended the usual RGB input to include beyond chromatic values. This allows us to use region-based properties as described above. Further use of other non-chromatic values will be investigated in the future.

*(D) Network architecture and convolution.* For our experiments, we chose a very basic network architecture. It consists of three convolutional layers and two dense layers with some dropout after each convolutional layer (Fig. S2). This is comparable to the LeNet model (*LeCun et al., 1989*) and simpler than other models that participated in the ImageNet competitions such as VGG (*Simonyan & Zisserman, 2015*) and AlexNet (*Krizhevsky, Sutskever & Hinton, 2017*). The reason for using a simple architecture is to investigate what kind of improvements we can confer over the simple network. The deeper network will offer more processing power and may obscure what improvements that we are investigating.

In 'Deep Learning Approaches to Texture' we discussed texture specific approach using Deep Learning. The common approach would use CNN to extract features and add on a separate classifier after. This classifier may be separate from the neural network. Otherwise the features are reduced by pooling and then they are connected to a fully connected network. In the average pooling, the pooling simply takes the average values of the feature map and use that as input to a dense network. As discussed earlier, convolution cannot extract certain features. Pooling also removes a lot of information some may be important. Our first improvement is to change input feature representation by preprocessing the input

pixels. The other innovation is the use of a convolution that is orthogonal to the spatial image. The convolution is applied over the feature channels rather than to the image plane as in normal convolution so that features, for the same spatial location, can be extracted directly across channels

Normal convolutions apply the kernel directly over the image to capture the image spatial characteristics. We might call this spatial convolution. We introduce a new style convolution that is applied in orthogonal channel depth-wise manner. We might call this orthogonal convolution. The kernel or filter is applied across channels (or feature maps) simultaneously (Fig. S3).

## EXPERIMENTS

This section verifies the accuracy and reliability of the proposed scheme through experimental results and comparison of the performance with several well-known deep neural networks.

### Orthogonal convolution with Haralick features on regional image patches

The objective is to test the impact for additional features on classification accuracy with a simple deep learning neural network.

In this experiment, we wanted to do a simple comparison on varying the input parameters against three different simple neural network architectures. The dataset we used is the OUTEX TC_00013 colour texture database. The dataset has 68 classes with 20 samples each, making a total of 1,360 image samples. The image size is 128 by 128 with RGB channels. We tested nine different input features against this one dataset. Training and Test split is 70:30. The Haralick feature used are as described in '(C) Sliding Window Calculation of GLCM Patches'.

The input features:

| | |
|---|---|
| **RGB** | The original three channel input without any changes |
| **H7** | Seven grayscale Haralick features are calculated on a sliding $5 \times 5$ window as described above. The input tensor would be $41 \times 41 \times 7$ |
| **RGB+H21** | Original size $128 \times 128 \times 3$ RGB and with $41 \times 41 \times 7$. H7 channel for each RGB (total $7 \times 3 = 21$) as 1 separate input branches. The other input is RGB layer. The two branches are flattened at the end of 3 convolutions, and then concatenated together before passing it on to the final two dense layer |
| **H42** | We calculate six different cross-colour circular GLCM, namely, GR, RB, GB, RR, BB and GG. Since each circular GLCM we extract seven Haralick's measure, these gives a total of $7 \times 6 = 42$ layers. The input tensor is $41 \times 41 \times 42$ |
| **H21** | This is similar to the above except we only use three cross-colour circular GLCM, GR, RB and GB. This gives $3 \times 7 = 21$ features for seven layers. The input tensor is $41 \times 41 \times 7$ |

| **RGB-H7** | One cross colour circular GLCM using GR and the original RGB texture image reduced to the same size of the extracted Haralick feature which is $41 \times 41$. The input tensor is $41 \times 41 \times 10$ |
|---|---|
| **RGB-H21** | Same as above except now we include more cross color circular GLCM pairs: GR, RB and GB. The original RGB resized to $41 \times 41$ is concatenated as additional 3 channels. The input tensor is $41 \times 41 \times 24$ |
| **RGB-H21-64** | The same as RGB+H21 except that the quantization level of the GLCM table has 64 levels instead of the default 16 levels |

There are three variant network architecture used with similar depth. The architecture that was used is basically similar to the Lenet architecture (*Sanchez et al., 2013*) which was one of the earliest network architecture.

In this experiment all the neural networks use orthogonal convolution. The first variant A1 uses non-separable convolution whereas A2 uses separable 2D convolution.

Separable convolutions compute convolutions using separate kernel filters on each input channel separately. This is followed by a pointwise convolution which mixes the resulting output channels.

Both A1 and A2 uses the non-conventional orthogonal convolution. The last network architecture A3 has separate branched inputs that are later flattened and concatenated.

## Experiment 2–normal convolution with Haralick features on regional image patches

We repeat the tests with the same features that were used in Experiment 1. However, we used the normal 2D convolutions that are applied on the image spatial dimensions. We also tested separable (A2) and non-separable convolutions (A1).

## Experiment 3–with other OUTEX dataset with orthogonal convolution

The first experiment was to determine which input features works best while keeping the network architecture relatively similar. The number of layers and architecture are kept the same. We tested on additional OUTEX datasets using the best feature and architecture obtain in Experiment 1. The datasets have different properties as below:

| | | | |
|---|---|---|---|
| *Outex 00013* | *20 per 68 class* | *1360 samples* | |
| *Outex 00034* | *20 per 68 class* | *1360 samples* | *different illuminants* |
| *Outex 00031* | *40 per 68 class* | *2720 samples* | *different resolution* |
| *Outex 00032* | *40 per 68 class* | *2720 samples* | *noise* |
| *Outex 00033* | *40 per 68 class* | *2720 samples* | *blur* |

## Experiment 4–with rotated OUTEX dataset 00030 with orthogonal convolution

This experiment tests how sensitive the features and network convolution are to rotations. We train the features and network on one rotation (zero rotation) then test it on a different

rotation ranging from 5 degrees to 90 degrees: 5, 10, 15, 30, 45, 60, 75 and 90 degrees of rotation. We train it on 1360 samples with 68 classes and test on 10 samples per class.

### Experiment 5–with rotated OUTEX dataset 00030 and normal spatial convolution

We repeat Experiment 4 with the commonly known VGG deep learning architecture.

## RESULTS

### Experiment 1

In the first experiment, we investigated the networks with various parameters, the most important determinants are the type of input features and convolution types. The results are presented in Table S2 and Fig. S4. We have found that separable convolution gave better results. Exact accuracy is not the main thrust of this research. The intention is show that there is an improvement with the introduction of different features other than RGB features.

We also found that more quantization levels do not produce better results. The quantization levels will capture better details for the image patch because more gray intensity levels are used but there is a possibility that it will over-fit. Surprisingly 16 gray levels are sufficient and is better than 64 levels (see Table S2; bottom two rows). The reason is that lower quantization level allows better pattern generalization with 16 levels being the sweet spot in this case.

The graphs at Fig. S5 show that using only either RGB (left) or Haralick measures (right) will overfit. This shows that either of each feature type is not sufficient to generalize the image texture patterns sufficiently. The training accuracy for both is close to 1.0 but the validation (test) accuracy is far below it.

The second pair of graphs in Fig. S6 show a better generalization curve for the validation (test) data. The training and validation (test) curve show a close match indicating that the training model is capable of generalizing to the test data. The first graph is derived from Haralick measures with cross colour pairs but with separable convolution. The second graph incorporates the RGB channels with the same Haralick cross colour pairs. The results indicate that the inclusion of RGB colour channels together with Haralick features improves the results.

The best results are obtained for RGB-H21-16 (16 quantized level). The confusion matrix for RGB-H21-16, which identifies where the classification errors are, shows that most errors occurred at the last few classes (according to the naming order of the classes). Visual inspections show that they are very difficult to distinguish from their visual pattern because they look very similar.

### Experiment 2

Experiment 2 repeats Experiment 1 but with normal spatial convolution. The results in Fig. S7 indicate that orthogonal convolution improves on the result of spatial convolution consistently.

### Experiment 3

Figure S8 and Table S3 show the results of our features and architecture against other well-known deep learning architecture such as the VGG-16. We show that our results are better consistently. The results obtained from other architecture are consistent with results reported in the literature (*Khaldi, Aiadi & Kherfi, 2019*).

### Experiment 4

The results (Fig. S9 and Table S4) indicate that our results are superior to VGG-16 architecture. Our approach is not as rotational sensitive as the VGG architecture.

## DISCUSSION

The results all combine to show that the orthogonal separable convolution with the Haralick features calculated over small regions are an improvement over other purely RGB-input based deep learning architecture. We specifically compared GoogleNet, ResNe, VggNet and AlexNet (see Table S5). Our neural network architecture is one with 3 layers of convolution, three pooling layers and two fully connected layers—it is one of the most basic architecture that is essentially the Lenet-5. Lenet-5 CNN (*LeCun et al., 1989*) that was first used successfully on digit recognition on the MNIST (Modified National Institute of Standards and Technology) dataset. Lenet-5 has seven layers that are made up of three convolutional layers, two subsampling layers and two fully connected layers. We present a comparison of our results against results reported (*Khaldi, Aiadi & Kherfi, 2019*) in Table S5.

For comparison, AlexNet (*Krizhevsky, Sutskever & Hinton, 2017*) has 5 convolution layer and three fully connected layers and VGG-19 (*Simonyan & Zisserman, 2015*) has 16 convolution layers and three fully connected layers. In addition, the GoogleNet (*Krizhevsky, Sutskever & Hinton, 2017*) has 22 layers made of nine inception layers. Each inception block has seven convolutions in four parallel branches. ResNet (*He et al., 2016*) have versions that are 50, 101 or 152 layers made up of multiple ResNet blocks.

Our network architecture has three convolution layers with 2 fully connected layers and the results obtained are better than any of the larger and deeper convolutional networks.

### Higher order statistics

The use of second- order measures has improved the results without the need to extend the depth of the neural network. The Haralick features are second-order measures of a second order measure. The GLCM histogram is a second order measure as it is a pixel-pixel histogram. The measures derived from the histogram are also of second order. These descriptors, being higher order statistics, are not easily calculated directly from raw RGB values using convolution, at least it will not be extracted with shallow networks. This calls into consideration the quality and type of input, and the notion that convolutional networks will be able to extract the appropriate features from the base RGB values. Our results show that shallow neural networks can be given a boost by the right type of input features especially those that can't be easily computed by convolution as higher order statistics can't be computed by a single convolution layer. The convolution layer basically

computes a linear combination of its input and passes that through a non-linear filter. It does not compute higher order statistics. Our results indicate that there are beneficial results if such input is fed into deep learning networks and better or comparable results can be obtained with shallower networks. This raises the question if the generalizations obtained by deeper networks are the same since the type of features used are different. This warrants further investigation. Our results suggest that we can expand the input beyond the RGB values to include other features including those traditionally used hand crafted or pre-computed features that could not be computed by convolution.

### Network overfit

*Basu et al. (2018)* provided an analysis indicating that the intrinsic dimension of texture dataset is higher than that can be shattered by the deep learning neural network. Their work showed that without explicit texture features, *i.e.,* using raw pixels, textures cannot be classified by deep neural networks because of their high intrinsic dimensionality. They also discussed that object datasets inherently have a lower intrinsic dimensionality compared to texture datasets. Their results show that classifying texture using texture-based features outperformed using raw pixels. They also showed that deeper networks like AlexNet performed worst on all datasets, indicating that such networks tend to over-fit.

### Orthogonal convolution

The orthogonal convolution detects patterns across feature vector values over a single spatial column (or row). It loses a portion of the spatial information. Only column (or row) spatial information is retained per convolution channel. However, it links to other features at the same location. Losing some spatial structural information may be beneficial for stationary textures as otherwise the deep learning network will encode local spatial structures that may not generalize well. Orthogonal convolution discards spatial information therefore it does not detect micro-structures well, rather it tries to correlate information at a local region but across its features.

The use of separable convolutions works better than non-separable convolutions. In the case of separable orthogonal convolution (Fig. S10), the first convolution is over the column and feature vector values, these are applied on a column-by-column basis which is then combined *via* the *1x1xn* convolution. The use of *1x1xn* convolutions after separable convolutions allows the channel depth layer to be selected independently, in our case across columns. The *1x1* convolution selects the resultant features from the first convolution.

### New network architecture

Results from previous results (*Khaldi, Aiadi & Kherfi, 2019*; *Geirhos et al., 2019*) indicates that deeper networks are adept in detecting microstructure type of texture but are relatively poorer with stationary textures. We can combine both approaches to create a best of both worlds approach, for example, by creating a double headed input network with one branch the conventional convolution architecture and the other branch with pre-calculated features.

## Limitations

This study does not address changes in input sizes and quality, especially when the same texture is presented in different sizes. The algorithm parameters will need to change for different sizes. We have not examined how to select these parameters automatically. Currently this study also does not address multiresolution images of the same texture. The other limitation is that the selected patch size is for a fixed input size making it not robust for different input size. The patch size is dependent on the input size and how it can be automatically selected is not addressed in this study. However, we examined the rotational invariance of the image textures, and our approach is robust to rotated textures.

The other limitation is that the orthogonal convolution discards most spatial relationships, so it is not useful for object detection or recognition since that requires recognizing spatial relationships.

## CONCLUSION

Deep learning inputs for computer vision are invariably direct pixel intensity values. These are not very useful for textures because textures are statistical distributions of pixel values. These distributions cannot be computed directly through convolutions or require more convolution to capture them. In this work, we employ two strategies to capture these statistical distributions.

Firstly, we divide the input image into patches instead of pixels to capture the local distribution variation by calculating the patch regional properties. In our case, we capture the grey level co-occurrence matrix (GLCM) for each patch. These GLCM properties cannot be calculated directly by convolutions.

Since our approach allows the use of pre-calculated features, these features could be replaced with any other features, not necessarily restricted to the ones that we have used (*Loke, 2018*). In the future work we will explore ways to incorporate other pre-calculated features. It could potentially include non-visual features and applicable to other domains other than computer vision, for example time-series data. Hence, this idea is not limited to just texture recognition and could have potential wider consequences. In our work we used fixed window sizes (patches) for calculating the regional properties. However, this need not be limited to fixed window sizes, the patch can be variable in size.

Second, we try to prevent the convolution from capturing local spatial patterns or microstructures by applying the convolution 90 degrees orthogonally to the spatial plane. This forces the convolution to discard learning spatial patterns. Our conjecture is that when the relationship between pixels can't be described by convolution, it will take many layers for the CNN to duplicate the effort. In our study we demonstrated experimentally, that for certain visual textures, spatially oriented convolution can't capture the relationships well at the shallow level.

Our novel approach has allowed a smaller network (less deep) to do a better job than a deeper network. Our network has only seven layers and it performs better than ResNet that has 50 layers.

We have proposed a new and viable convolution architecture that is suitable for detecting texture. We also showed that it is less sensitive to rotation because of the structure of the

convolution. This convolution, orthogonal as opposed to the spatial approach, can be generalized to include other orthogonal convolutions (Fig. S11). Additionally, it is also possible to investigate convolutions at different angles. All these convolutions can capture different feature relationships.

In summary, we have introduced a new way of utilizing convolution so that different feature relationships can be captured. This significantly broadens the type of relationships or correlations that can be captured. Finally, we introduced regional calculated properties as input instead of pixel values. These regions can be of variable sizes.

### Funding
The author received no funding for this work.

### Competing Interests
The author declares there are no competing interests.

### Author Contributions
- Kar-Seng Loke conceived and designed the experiments, performed the experiments, analyzed the data, performed the computation work, prepared figures and/or tables, authored or reviewed drafts of the article, and approved the final draft.

### Data Availability
The Extended Outex texture classification test suites are available at: https://color.univ-lille.fr/datasets/extended-outex/.

The code is available at GitHub and Zenodo:

- https://github.com/ksloke/texture

- ksloke. (2023). ksloke/texture: v1.0.0 (v1.0.0). Zenodo. https://doi.org/10.5281/zenodo.8378950.

### Supplemental Information
Supplemental information for this article can be found online at http://dx.doi.org/10.7717/peerj-cs.1927#supplemental-information.

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
