# Peer review of "A novel approach to texture recognition combining deep learning orthogonal convolution with regional input features"

_PeerJ Computer Science, doi:10.7717/peerj-cs.1927_

## Round 0.1 · original submission · Major Revisions

Based on the referee reports, I recommend a major revision of the manuscript. The author should improve the manuscript, taking carefully into account the comments of the reviewers in the reports, and resubmit the paper.

**Language Note:** The review process has identified that the English language must be improved. PeerJ can provide language editing services - please contact us at copyediting@peerj.com for pricing (be sure to provide your manuscript number and title). Alternatively, you should make your own arrangements to improve the language quality and provide details in your response letter. – PeerJ Staff

Reviewer 1 ·

Basic reporting

This paper claims that it advances the convolutional neural network (CNN) -based texture representation. This is achieved by encoding the local neighborhood of the input images with texture information while feeding into a CNN. Few experimental results are provided to support their contribution.

Experimental design

No comment

Validity of the findings

Results should be given in a tabular form. How much performance is improved with respect to traditional CNN is not mentioned.

Additional comments

Need extensive revision, English flow improvement, and result presentation.

·

Basic reporting

The paper is well explained with cited references and good grammatical English. The abstract provides a clear overview of the research topic, stating the significance of textures for segmentation and object detection. It acknowledges the success of deep convolutional nets like VGG and ResNet in non-stationary texture datasets, while highlighting the limitations of deep networks in detecting stationary textures.

The abstract also introduces the proposed approach of using simple convolutional networks with pre-calculated regional features, resulting in improved accuracy compared to popular deep network architectures. Overall, the abstract effectively conveys the research objectives and findings.

Literature review is well done and study of existing work is well explained with relevant articles. Missing reference citations in the text. The manuscript mentions a reference [30-43], but no corresponding citation is provided. Please ensure that all references mentioned in the text are properly cited to support the information and claims made in the manuscript.

Figures are provided with better clarity.

Experimental design

1.The abstract will be more benefit if provided with more specific information about the results and findings. While it mentions an average improvement of 8.5% in accuracy mentioned in Line 25 over various deep network architectures, it would be helpful to highlight any significant observations or comparisons made in the study.
2.The evaluation metrics used to measure accuracy is mentioned throughout the article. If necessary, providing information about the specific metrics employed (e.g., precision, recall, F1-score) would enhance the clarity and completeness of the abstract. in terms of performance
3.In Feature Representation section 3.2 line number 245, the authors could further explain the process of calculating GLCM on patches instead of globally.
4. Line 405: The manuscript should provide more details about the experimental design. Information such as the dataset used, sample size, training/validation/test splits, and any preprocessing steps would enhance the reproducibility of the study.
5. Line 415: The statement that "surprisingly 16 gray levels are sufficient" should be supported with evidence or statistical analysis. If possible, provide a rationale for choosing 16 gray levels and compare it to the results obtained with other levels.
6. Line 439: The manuscript refers to difficult classes in Figure 9 but does not provide any description or analysis of these classes. It would be helpful to include a discussion on the specific challenges posed by these classes and potential reasons for the classification errors

Validity of the findings

1..The results of Experiment 1 are presented in Table 2 and Figures 6-8.
2.The performance of different input types and architectures is analyzed, with separable convolution showing better results.
3.The impact of quantization levels on the accuracy is discussed with the dataset Outex images.
4.Figures 7 and 8 illustrate the training and validation accuracy for different feature combinations.
5. The best results are reported for RGB-H21-16, and a confusion matrix is mentioned, but it would be helpful to provide more details or insights from the confusion matrix analysis.
6.Experiment 2 is briefly mentioned with Figure 10, but further discussion or interpretation of the results would be valuable. Conclusion shall reflect the results obtained as a summary.
7.Line 468: The manuscript claims that the proposed approach outperforms other deep learning architectures, but no comparative analysis or statistical tests are provided. It would be beneficial to include a detailed comparison of performance metrics (e.g., accuracy, precision, recall) between the proposed approach and other architectures to support this claim.
8.Line 470: The text states that the proposed approach is an improvement over "other purely RGB deep learning architecture," but it lacks clarity on which specific architectures are being compared. It would be helpful to specify the compared architectures and provide evidence or references to support the claim

Additional comments

Overall, the paper provides a good foundation for the proposed scheme and presents relevant experimental results

·

Basic reporting

This paper is an excellent and experimental summary of how simple convolution networks can obtain better results than deep networks. Material is understandable. There are considerable technical jargons that need to be simplified wherever appropriate for non-expert readers. While mentioning technical discussion, wherever applicable sentences / paragraphs need to be restructured and logical transitions should be used to improve readability. Figures and tables should be added where they are being discussed, not just collectively at the end making it challenging to follow. The article covers a lot of good content.

Experimental design

The study fits in scope of the journal. However according to the PeerJ standards, submissions should clearly define the research question. There is no "research question" in a classical sense or gap in the research that is being addressed. It seems there is not enough information to replicate this study, dataset is not attached. Orthogonal Convolution architecture is insufficiently discussed. Methods and Approach are described in good details. Article has good findings, it fits more in criteria of “experimental findings” or “experimental study” than “research investigation”. Good efforts.

Validity of the findings

This article is missing information why idea of simple convolution network is novel in this study of stationary textures. The conclusions must be appropriately stated and should be connected to the original question that is being investigated and should be limited to what is supported by the results. There is not much discussion about - the limitation of this findings, any improvement suggestions in approach taken and any future scope of the work.

Additional comments

Abstract:

The abstract is clear and concise it could be improved further.
1. Specify what makes this technique novel and explain how regional features were pre-calculated.
2. Clarify if the 8.5% improvement in accuracy is over a specific dataset.
3. Explain if there are particular applications where the detection of stationary textures is important.
4. Compare the proposed model to other models focusing on stationary textures.

Introduction:

The introduction is well-written and informative. The author does a good job of summarizing the relevant literature and the key concepts. Further could be added.
1. Consider adding a formal definition for textures.
2. Explain why studying textures is important and provide practical applications or the need for detecting stationary textures.
3. Provide a brief explanation of "Population Modulation Index (PMI)" , "spectrally matched (SM) and correlation matched (CM) textures" for non-expert readers.
4. Briefly mention current methods focusing on stationary textures and explain why they are inadequate. Highlight the gap in existing research that needs addressing.
5. Restructure sentences for smoother transitions and logical flow of ideas. Ex. For instance, paragraph on line 72 about deep nets seems to be a bit out of place and could be brought forward when discussing CNNs. Another example line 57 “CNN layers after L1 respond substantially better than lower layers” – this could be restructured as “layers after L1 respond substantially better to textures than the lower L1 layer”
6. Group similar ideas together and use transition sentences to enhance the flow. Ex. you could separate discussions of previous research, definitions of texture, and texture detection in the brain into different paragraphs for improved readability.
7. Clearly state the problem statement and have a separate concise paragraph to address the issue or problem that this research intends to solve.
8. Explain the benefits and costs of using the mentioned classifiers, K-nearest neighbor and SVM.

Materials & Methods - Related Works - 2.1 Features for Texture:

This section provides a good explanation of texture analysis, including various techniques and their methodologies. Few areas that could be improved are as below.
1. Provide more clarification and explanation about concepts such as textons, the significance of texture analysis, challenges in texture analysis, and the structural and statistical approaches.
2. Define micro-structures and chi-squared statistic with simple definitions and explain their significance.
3. Add more brief details about the LBP method.
4. Explain the importance of the specific approaches used in texture analysis and highlight their benefits.
5. Provide strengths and weaknesses of each method and indicate the most applicable areas for each method.
6. Consider adding diagrams or figures to illustrate the various methods of texture analysis.
7. Clearly mention if any method or methods that are being discussed here are used in this study.
8. Conclude this section with a summary including key points.

Materials & Methods - Related Works - 2.2 Deep Learning Approaches to Texture:

This section has great discussion regarding multiple approaches towards texture recognition using Convolutional Neural Networks (CNNs). It provides a clear overview of the latest studies. For more information content, consider adding below.
1. Create a summary table that directly compares the performance and computational cost of the different methods mentioned. Explain why certain techniques performed better or worse, what differentiates them, and how they can be improved.
2. Provide explanations for terms such as "Fisher Vectors," "FV-CNN," "VLAD," "residual encoding layer," or "bilinear pooling" for non-expert readers.
3. Briefly explain key concepts like pooling, feature extraction, and feature maps.
4. Include more details on different deep learning architectures used for texture classification.
5. Discuss the challenges of deep learning texture classification, such as the need for large training datasets and sensitivity to noise.
6. Explore future directions for deep learning texture classification, such as the use of attention mechanisms and the development of new loss functions.
7. Discuss other ways to improve the performance of deep learning models, such as transfer learning for texture classification.
8. Address multi-modal approaches for texture classification, considering the use of multiple sources of information.
9. Highlight domains or real-world applications where texture recognition is important.
10. Mention limitations of current deep learning techniques for texture recognition and briefly discuss areas that future research needs to address.
11. Proofread the section for minor grammatical errors and simplify sentence structures. For example, change "deep learning approaches generally uses the CNN layers" to "deep learning approaches generally utilize CNN layers."

Methods and Approach:

1. Clarify the descriptions of technical procedures such as the calculation of GLCM and the Haralick features.
2. Include figures and tables in the relevant sections instead of placing them all at the end.
3. When referring to "Image Size," specify whether it refers to spatial dimensions or image resolution.
4. Provide the rationale behind selecting the specific network architecture used in the study and mention if other comparable architectures were considered.
5. Provide more explanation and or interpretation of the mathematical formulas used, demonstrating how each formula is applied to the data.
6. Discuss the rationale behind using orthogonal convolution over other convolution types in deep learning.

Discussion:

1. Specify the datasets used in the experiments to aid understanding of the results.
2. Discuss any limitations of the approach and address whether the output varies with changes in the size and quality of input images.
3. Suggest ways to improve the performance of the approach or extend it to other types of texture classification problems.
4. Clarify if there are any specific Haralick features used.

Conclusion:

1. Conclusion should include key findings of the research conducted. It is an opportunity to mention significance and advantage of the proposed approach. This needs to be added.
2. Are there any domains where future study would benefit?
3. In the context of texture analysis or computer vision, non-visual features are not part of pre-calculated features.
4. Briefly mention what other pre-calculated features could be added for further studies?

---

## Round 0.2 · Minor Revisions

Reviewers are satisfied with the revised version of the manuscript. Thus I'm provisionally accepting the manuscript. Kindly revise the manuscript as per the reviewer's suggestions and resubmit it.

·

Basic reporting

No Comments

Experimental design

No Comments

Validity of the findings

No Comments

Additional comments

• Satisfied to note that all of the review comments have been addressed thoroughly and to satisfaction. The author's responses have clarified the issues raised, and they are acceptable.
• References not cited in text have been removed.
• Figures are provided with better clarity.
In Experimental design,
 Understood that the availability of metrics in the literature and datasets can pose challenges.

·

Basic reporting

Happy with version 2 , refer Additional comments for minor revision.

Experimental design

Happy with version 2 , refer Additional comments for minor revision.

Validity of the findings

Happy with version 2 , refer Additional comments for minor revision.

Additional comments

Thank you very much for addressing the comments and happy with this version. Having reviewed version 1, I can tell this version 2 has much smooth flow now, easier to read and is grouped logically. Thank you for sharing your concerns regarding the suggested revisions. I understand your reservations about the article length and scope. However, I'd like to emphasize the importance of clarifying the context of the proposed approach by briefly mentioning the different deep learning architectures used for texture classification. This is a peer review #4 on section "Materials & Methods "

My intention behind this comment is not to have an exhaustive description of each architecture but rather a succinct overview or comparison, highlighting the distinctiveness and relevance of the proposed approach. A few well-chosen sentences or a compact table could effectively convey this information without significantly increasing the article length. While I appreciate the references provided, expecting readers to refer to external sources for basic contextual information can make the article less self-contained and possibly reduce its impact. This results in less citations for your important work. Briefly touching upon the existing architectures would provide a more comprehensive understanding for readers unfamiliar with the nuances of texture classification architectures.

I'd like to stress the significance of the comment regarding discussing the challenges of deep learning for texture classification. This is #5 on section "Materials & Methods " . Including a brief section or a few sentences addressing the known challenges doesn't necessarily mean explaining prior work in-depth or re-implementing those works. Rather, highlighting the common challenges would:

(1) Set a clear context for readers on why a novel approach like the one you're proposing might be necessary.
(2) Improve the article's clarity by highlighting the issues you seek to address .
(3) Provide a more holistic view of the texture classification landscape, ensuring that both newcomers to the field and seasoned experts appreciate the significance of your work.

While I acknowledge the provided references on review articles, the readers might benefit from a concise in-article overview of these challenges, especially when considering the relevance of your novel approach. I kindly urge you to reconsider this feedback. I am sending with minor revisions. If you decide to address, it does not need another review from me. This article is good.

This is a amazing work, I enjoyed reading version 2 and no doubt this would add value to the current texture classification deep learning approaches. Thank you very much again for this great study with lot of important details. Thank you for the opportunity to review.

---

## Round 0.3 · accepted · Accept

The author has addressed the reviewers' comments properly. Thus I recommend publication of the manuscript.